# Two-Factor-Based Public Data Protection Scheme in Smart Ocean Management

**DOI:** 10.3390/s19010129

**Published:** 2019-01-02

**Authors:** Jian Shen, Xinzhao Jiang, Youngju Cho, Dengzhi Liu, Tianqi Zhou

**Affiliations:** 1Jiangsu Engineering Center of Network Monitoring, Nanjing University of Information Science & Technology, Nanjing 210044, China; s_shenjian@126.com (J.S.); jxz942@163.com (X.J.); liudzdh@126.com (D.L.); tq_zhou@126.com (T.Z.); 2State Key Laboratory of Cryptology, Beijing 100878, China; 3Guangxi Key Laboratory of Cryptography and Information Security, Guilin 541004, China; 4SW Convergence Education Institute, Chosun University, Gwanju 61452, Korea

**Keywords:** two-factor, public data protection, redistribution, attribute-based cryptography, smart ocean management

## Abstract

Nowadays, two-factor data security protection has become a research hotspot in smart ocean management. With the increasing popularity of smart ocean management, how to achieve the two-factor protection of public data resources in smart ocean management is a serious problem to be tackled. Furthermore, how to achieve both security and revocation is also a challenge for two-factor protection. In this paper, we propose a two-factor-based protection scheme with factor revocation in smart ocean management. The proposed scheme allows data owners (DOs) to send encrypted messages to users through a shipboard server (SS). The DOs are required to formulate access policy and perform attribute-based encryption on messages. In order to decrypt, the users need to possess two factors. The first factor is the user’s secret key. The second factor is security equipment, which is a sensor card in smart ocean system. The ciphertext can be decrypted if and only if the user gathers the key and the security equipment at the same time. What is more, once the security equipment is lost, the equipment can be revoked and a new one is redistributed to the users. The theoretical analysis and experiment results indeed indicate the security, efficiency, and practicality of our scheme.

## 1. Introduction

The construction of smart ocean management has become an important trend of smart ocean field. With the increasing requirements for the quality of management, a large number of problems have emerged. On the one hand, it is very significant to structure a secure memory space to ensure the data security (nautical data, ship position data, and ocean resources data, etc.). On the other hand, it is a problem that how to access these data safely and efficiently to meet the needs of navigation department, radar department, electromechanical department, and other departments on the warship. Especially, it has become an urgent problem that access to public data, such as computer rooms, conference rooms, control rooms, and other public resources. However, it is difficult to solve such thorny problems with traditional data processing schemes. Therefore, information and communication technologies to tackle the problem of smart ocean management emerge at a historic moment [1,2].

Because smart ocean management involves a large amount of complex data, for instance, confidential navigation roadmaps, high-dimensional remote sensing satellite image, and marine resource distribution maps, therefore, cloud computing technology is usually applied to shipboard server (SS). Cloud computing is an innovative change in smart ocean management. Compared with traditional computing technology, cloud computing has a lot of advantages in terms of storage and calculation of ocean data, therefore, it has received extensive interest from the academic community [3,4,5]. By using cloud computing technology in smart ocean management, users can not only get limitless storage space, but also get limitless computing resources [6]. Server data storage [7,8,9] is a branch of distributed storage model, which is one of the most significant applications in smart ocean management. SS storage has many advantages, and the most extensive thing is data availability. However, the research on public data access is not too comprehensive, which refers to the situation where many users access public data. An occasion is smart ocean management as mentioned above. When acquiring important public resources, for example, crew and captain read confidential documents, security equipment and secret keys (two factors) are essential for users. Security equipment and secret keys are distributed by the central authority (CA) which communicates with the ship through a secure node transmission channel.

To prevent data from being stolen, researchers mostly design single public key encryption schemes [10,11,12]. In general public key encryption schemes, the secret key required for decryption is often related to the public key. The key is generally stored in a private device or a trusted third party (TTP). If there is no malicious attack, this kind of secret key storage scheme is secure enough. Unfortunately, it often goes against one’s wishes. When being accessed through the Internet, private devices, and TTP are most likely to be attacked by illegal hackers, resulting in the secret key being stolen. But for all these, the users know nothing. In addition, please consider the following two real-life work occasions: Personal computers that store a user’s secret key may be used by others in cockpits, laboratories, and offices, public computers that record users’ login information will be shared by different users. Under these circumstances, the secret keys are able to be compromised by some malicious attackers who use technical or non technical means. Therefore, single secret key encryption scheme no longer satisfies certain security requirements in smart ocean management, and the two-factor data protection scheme arises at the historic moment.

We note that there are some other research works on two factors, such as [13]. Nevertheless, the two factors in their works are different from ours, they leverage different technologies to design different systems. Here, we will not compare them with our present scheme.

As far as we know, ref. [14], for the first time, provides two-factor data protection to support security device revocability. While the shceme in [14] is actually an identity-based scheme, and ciphertext can be decrypted by only one user rather than a group of users. Therefore, ref. [14] is only a solution to the storage problem of single user data, but it is not suitable for protecting public data in smart ocean management. In the era of shared economy, it is more worthwhile to study the privacy protection of public data.

### 1.1. Our Contribution

Inspired by [14], we propose a two-factor public data protection scheme in smart ocean management. Note that message validation is presented to verify the correctness of the message after decrypting. The contributions of our research are as follows:A practical attribute-based encryption (ABE) data protection scheme is proposed. In practical applications, public resources are more likely to be shared by many users. The security of these public resources is of great significance. In our scheme, we take advantage of an efficient ABE to address this problem. Ensure that only users who satisfy specific attributes can access public data.A two-factor-based data protection scheme that supports revocation in smart ocean management is proposed. We apply this kind of data protection scheme to smart ocean management for the first time. Ocean-related data is often highly confidential, however, a single factor scheme does not meet the security requirements of the application layer. Therefore, we design a two-factor-based data protection scheme. In addition, when performing missions in the ocean environment, the sensor cards of the crew will inevitably be lost, for example slipping into the sea. Here, the revocation of the sensor cards is particularly important. Therefore, the revocation and redistribution of security equipment is also supported by our scheme.A scheme that security equipment is independent of secret key is proposed. In our scheme, the distribution and update of security equipment is separated from secret key. Therefore, the threat to secret key due to the loss of the security equipment can be effectively reduced.A SS security scheme is proposed. As is known to all, SS is a semi-trusted entity. However, in our scheme, SS cannot decrypt any ciphertext. At the same time, users can also complete the correctness verification of decrypted message.More formal and complete security and performance analysis are proposed. According to the designed security model, descriptive language and rigorous mathematical proof are used in security analysis. The attacker’s success is reduced to the resolution of difficult problems. The security of the scheme is proved probabilistically. In performance analysis, similar schemes are compared with ours from different phases, thus achieving a more fine-grained comparison.

### 1.2. Related Works

In recent years, emerging technologies are booming and are reflected in many fields. In particular, the application and development of space and ocean. In 2018, Kim and Ben-Othman [15] proposed a surveillance model for multi domain IoT environment, which is supported by reinforced barriers with collision-avoidance using heterogeneous smart unmanned aerial vehicles. In the field of oceans, in order to solve some problems in data collection of underwater acoustic sensor networks (UASNS), Han et al. [16] proposed a stratification-based data collection scheme for three-dimensional UASNs. In addition, to achieve accurate and energy efficient trust evaluation in UASNs, an attack-resistant trust model based on multidimensional trust metrics is also proposed by Han et al. [17].

In addition to the research directions mentioned above, research on SS also attracts the attention of academic groups at home and abroad. Among them, storage is a major research hotspot in SS. Various sub fields of storage, for instance, data encryption [18], location detection [19], privacy protection, and data sharing [20,21] are the subjects of intense discussion as well.

In 2005, Sahai and Waters [22] first put forward fuzzy identity-based encryption (IBE) and further discussed in [23], which are the original work of ABE. After that, two variants of ABE were proposed. They are key-policy ABE (KP-ABE) [24] and ciphertext-policy ABE (CP-ABE) [25,26] respectively. The difference between them is that a given policy is associated with a key or a ciphertext. While the CP-ABE is the opposite of KP-ABE, and CP-ABE has a more important practical value. Later, lots of CP-ABE schemes with specific features are introduced by researchers. In 2008, Boldyreva et al. [27] proposed an identity-based encryption scheme that supports efficient revocation operation. Now, the scheme has been applied in KP-ABE. In 2010, Yu et al. [28] provided a CP-ABE scheme with attribute revocation. However, the length of private key and ciphertext are positively related to the number of required attributes. In addition, all attributes must be involved in key generation, encryption and decryption. Therefore, the scheme greatly increases overhead of computing and communication. Of course, in addition to revocability, there are many ABE scheme with other features. For example, schemes [24,25,26,29,30,31,32] that require a fully trusted authority. Nevertheless, by leveraging above schemes can only achieve access control but not two-factor data protection, let alone revocation of factor in public data occasion.

In 2017, Shivanna et al. [33] presented a double encryption privacy protection scheme. However, double encryption has a lot of defects. Loss of security equipment causes ciphertext to never be decrypted. Furthermore, this scheme makes the encryption process more complicated. In 2002, Dodis et al. [34] first designed a key-insulated public key scheme to solve the problem of private key exposure. In 2003, Dodis et al. [35] applied key insulated technology to digital signatures. However, in [34], there is a great correlation between the master key and the public key. What is more, in [35], frequent update of private key may lead to compromise of master secret key. For enhancing security of the master key, in 2006, Hanaoka et al. [36] introduced the parallel key insulated public key encryption scheme. But the security of [36] is proved under random oracle model. In 2007, Quisquater et al. [37] proposed a parallel key-insulated public key encryption scheme in the standard model. In 2016, Wang et al. [38] redesigned attribute-based data sharing mechanism to solve the key escrow problem. Simultaneously, weighted attributes in access policies are constructed by [38] to improve the expression of scheme. However, the two-factor feature is not supported by [38], let alone equipment revocation. In most of the above schemes, update of user’s private key requires the participation of security equipment. This is obviously not suitable for the protection of public data. Because we hope that the user’s private key is updated occasionally, and the security equipment is separated from the private key.

In contrast to the above schemes, we design a two-factor-based public data protection scheme in smart ocean management by utilizing an efficient ABE and public key encryption. Importantly, compared with [14], the proposed scheme can achieve protection of public resources, not just personal data.

### 1.3. Organization

The rest of this paper is made up of the following sections. Section 2 introduces some preliminary knowledge in cryptographic so that make it easier for readers to understand our works. The system model, notations, security model and system components are illustrated in Section 3. Section 4 describes our scheme in detail. Section 5 and Section 6 show the security and performance analyses, respectively. Finally, Section 7 summarizes this paper.

## 2. Preliminaries

Before introducing our scheme in detail, it is inevitably to introduce some preliminaries in this section, including the bilinear maps, computational assumption, and attribute-based encryption.

### 2.1. Bilinear Maps

First, we define an algorithm, then input a security parameter *k*. Then, the algorithm outputs some parameters related to bilinear map, that is $\left(q,g_{1},g_{2},G_{1},G_{2},e\right)$, where $G_{1}$,$G_{2}$ are two multiplicative cyclic groups with prime order $q\in \Theta (2^{k})$ and $g_{1}$,$g_{2}$ are generator of $G_{1}$. A bilinear maps *e*: $G_{1}^{2}\to G_{2}$ is efficient map when it satisfies the following three properties:

Bilinearity. For all $g_{1}$, $g_{2}$
$\in G_{1}$ and $a,b\in_{R}Z_{q}^{*}$, $e\left(g_{1}^{a},g_{2}^{b}\right)=e{\left(g_{1},g_{2}\right)}^{ab}$;

Non-degeneracy. $\exists g_{1}$, $g_{2}$
$\in G_{1}$, there is $e\left(g_{1},g_{2}\right)\neq 1_{G_{2}}$. Where $1_{G_{2}}$ represents the unit element of $G_{2}$;

Computability. $\forall g_{1}$, $g_{2}$
$\in G_{1}$, there is at least an efficient algorithm to compute $e(g_{1}.g_{2})$.

### 2.2. Computational Assumption

*q*-weak Decision Bilinear Diffie-Hellman Inversion (*q*-wDBDHI) Assumption. For an algorithm A, the advantage of A decides the *q*-wDBDHI is ξ. For the following equation:(1)$$\begin{array}{c} \left|Pr[A\left(g,g^{a},\ldots ,g^{a^{q}},g^{b},e{\left(g,g\right)}^{b/a}\right)=1\right]\\ -Pr[A\left(g,g^{a},\ldots ,g^{a^{q}},g^{b},e{\left(g,g\right)}^{z}\right)=1]|=\xi \end{array}$$
there $a,b,z\in_{R}Z_{q}^{*}$. *q*-wDBDHI assumption holds when ξ is negligible for any polynomial time algorithm.

### 2.3. Ciphertext-Policy Attribute-Based Encryption

CP-ABE is a cryptography technology for realizing one to many secure communication, where the DO shares message to specific users by constructing an access policy and embedding the policy into ciphertext. The most primitive CP-ABE consists of following four algorithms.

$Setup(1^{k}).$ This algorithm takes a security parameter *k* as input. It outputs a public parameter PK and a master key MK.

KeyGen(PK,MK,S). This algorithm takes public parameter PK, the master key MK and an attribute set *S* as input. It outputs a private key SK related to the attribute *S*.

Encrypt(PK,M,A). This algorithm takes the public parameter PK, a message *M* and an access policy A as input. It outputs ciphertext CT such that only the user whose attribute set satisfies the access policy can decrypt.

Decrypt(PK,CT,SK). This algorithm takes the public parameter PK, a ciphertext CT and a private key SK as input. If and only if the attribute set *S* of the user satisfies the access policy A, the algorithm can decrypt the message *M* successfully.

## 3. Problem Statement

### 3.1. The System Model

As described in Figure 1, the whole model contains four entities: Central Authority (CA), Data Owners (DOs), Users and Shipboard Server (SS).

Central Authority: A CA is considered to be a entity that possesses unlimited computing and storage capacity. Meanwhile, a CA is also a trusted party, and its tasks are to generate system parameters, manage users (i.e., enrolling users: distribuing the secret key to every user) and distribute security equipment (sensor cards). Furthermore, the update of the security equipment is also responsible for CA. In the process of updating, CA redistributes a security equipment to the user and informs SS to update the ciphertext. Figure 2 shows the process of update.Data Owners: DOs are owners of message stored in SS. All the message is encrypted by using ABE. Finally, DOs upload the generated ciphertext to SS.Users: In smart ocean management, sailors, helmsman, managers and other crew are users. They can download the encrypted public data in SS. If the users want to get the message, they firstly do decrypt by using their security equipment and obtain the resulting primary ciphertext, then users with specific attributes can decrypt primary ciphertext by using their secret keys.Shipboard Server (SS): It is not a credible entity in smart ocean management. Concretely, SS is honest-but-curious, which can honestly implement the assigned tasks and return corresponding results. However, it will also do its best to collect sensitive information. Generally, SS is regarded as a party with unlimited computing power and storage space. In this paper, the DOs upload the encrypted message (primary ciphertext) to SS, then SS uses the public information obtained from CA to encrypt primary ciphertext, resulting in secondary ciphertex. In addition, SS is responsible for updating ciphertext when users’ security equipment is redistributed.

### 3.2. Notations

As is shown in Table 1, some primary notations used in our scheme are listed.

### 3.3. Security Model

In this paper, from the perspective of two factors, we mainly consider the following two threat models:Type-1: Decrypt without security equipment. In this case, the adversary has the right secret key, however, it has no security equipment, or security equipment and secret keys do not match.Type-2: Decrypt without secret key. This attack model is opposite to the previous model. In this situation, the adversary has a security equipment but no secret key, then it tries to decrypt the ciphertext.

### 3.4. System Components

The two-factor-based public data protection scheme in smart ocean management consists of six algorithms. The six algorithms are described separately as follows.

Setup: ($1^{k}$)⟶(param, msk). The algorithm is run by CA. A security parameter *k* is taken as input. The algorithm outputs public parameters param and master key msk.

Keygen and Security Equipment Distribution: (param, msk, *P*)⟶($sk_{P}$, $epk_{i}$, $esk_{i}$). The algorithm is run by CA. On inputting the public parameters param, the master secret key msk and the attribute *P* that users possess, the algorithm outputs secret key $sk_{P}$, public information $epk_{i}$, and secret information $esk_{i}$ of security equipment.

Primary Encryption: (param, *A*, *m*)⟶($C_{1}$). The algorithm is run by DOs. The input includes the public parameters param, the message *m* and attribute set A. The output is the primary ciphertext $C_{1}$.

Secondary Encryption: (param, $epk_{i}$, $C_{1}$)⟶($C_{2}$). The algorithm is run by SS. The public parameters, public information $epk_{i}$ of security equipment and primary ciphertext $C_{1}$ are taken as input. The algorithm outputs secondary ciphertext $C_{2}$.

Security Equipment Redistribution and Ciphertext Update: (param, $epk_{i}$)⟶($C_{2}^{*}$). The algorithm is run by CA and SS. On inputting the public parameters param and $epk_{i}$, the algorithm outputs the ciphertext $C_{2}^{*}$.

Message Decryption: ($esk_{i}$, $sk_{P}$, $C_{2}$ or $C_{2}^{*}$)⟶(*m*). The algorithm is run by users. The input includes secret information $esk_{i}$ of security equipment, secret key $sk_{P}$ and secondary ciphertext $C_{2}$ or $C_{2}^{*}$. The output is message *m*.

## 4. Our Scheme

### 4.1. Setup

All public parameters and master key will be generated in the setup phase. These public parameters will be shared among all parties (including DOs, Users, SS, and CA). However, the master key can only be kept by CA. The specific process of setup is as follows.

–We define $G_{1}$ and $G_{2}$ as cyclic multiplicative groups of prime order *p*, and *e*: $G_{1}^{2}\to G_{2}$ is the bilinear map.–Choose $g,g_{2},h\in G_{1}$, $\alpha ,\beta \in_{R}Z_{q}^{*}$. Here *k* is a security parameter. Four collision resistant hash functions are chosen as follows: $H_{1}:G_{1}\to Z_{q}^{*}$, $H_{2}:{\left\{0,1\right\}}^{*}\to Z_{q}^{*}$, $H_{3}:G_{2}\to {\left\{0,1\right\}}^{*}$ and $H_{4}:{\left\{0,1\right\}}^{*}\to G_{1}$. Meanwhile, setting $g_{1}=g^{\alpha }$.–There are *n* attributes in our scheme. The attribute set can be denoted as $A={\left\{A_{1},A_{2},\ldots ,A_{i},\ldots ,A_{n}\right\}}_{1\leq i\leq n}$. Each attribute $A_{i}$ has multiple attribute values $V={\left\{v_{1},v_{2},\ldots ,v_{i},\ldots ,v_{m}\right\}}_{1\leq i\leq m}$. Each attribute value can be used as a user’s ID, but the pre *N* bits of the attribute value is used as a common attribute of the user, here *N* is a threshold. Give a simple example, 201801234 is a crew number of a crew, that is, the ID of the crew. 2018 is the crew’s year of admission, 01 is a department number, 234 is the sort number of the crew. When *N* is equal to 4, crew enrolled in 2018 can be identified. When *N* equals 6, the department can be identified. The public parameters param is set to be $\left(k,q,g,g_{1},g_{2},h,e\left(g,g\right),H_{1},H_{2},H_{3},H_{4}\right)$.

### 4.2. Keygen and Security Equipment Distribution

Firstly, CA will distribute security equipment for every user according to the their $ID_{i}$. Secondly, CA is responsible for generating the secret keys for users which have specific attributes. Users can use their own security equipment and secret keys to decrypt ciphertext. The specific process is as follows.
–The CA chooses $z_{i}\in_{R}Z_{q}^{*}$, and sets the public information of the security equipment as $epk_{i}=g^{z_{i}}$, and its corresponding secret information as $esk_{i}=z_{i}$. Finally, CA distributes a security equipment to a user $ID_{i}$ and shares $\left(epk_{i},ID_{i}\right)$ with the SS.–CA computes
(2)$$\begin{array}{c} \tau_{i}=H_{4}{\left(s\right)}^{-H_{2}\left(\beta ||i\right)},\upsilon_{i}=H_{4}{\left(s\right)}^{-H_{2}\left(\alpha ||i\right)} \end{array}$$The secret key is $sk_{P}=\left(s,\tau_{i},\upsilon_{i}\right)$, where *s* is a mapping of the user’s attributes to strings. In addition, the set of attributes of each user is mapped to a unique string. *P* is a set of user-owned attributes.

### 4.3. Primary Encryption

DOs encrypt message based on attributes and send the encrypted messages to SS. Know public parameters param, message $m\in {\left\{0,1\right\}}^{*}$ and attributes set *A*. The process of primary encryption is as follows.
–Compute $c_{1}=m\cdot \alpha_{A}^{k}$, $c_{2}=g^{k}$, $c_{3}=\beta_{A}^{k}$, $c_{4}=A$, $M=H_{4}\left(m\right)$, and define $\alpha_{A}=\prod \alpha_{i}$, $\beta_{A}=\prod \beta_{i}$. Send the primary ciphertext $C_{1}=\left\{c_{1},c_{2},c_{3},c_{4}\right\}$ to SS and broadcast *M* to all users.

### 4.4. Secondary Encryption

After receiving the primary ciphertext from DOs, SS will encrypt it second times, resulting in secondary ciphertext. Knowing public parameters param, a primary ciphertext for the user and the information $epk_{i}$. The SS encrypts $C_{1}=\left\{c_{1},c_{2},c_{3},c_{4}\right\}$ to secondary ciphertext as follows
–Choose $\mu_{1},\mu_{2}\in_{R}{\left\{0,1\right\}}^{*}$, set $r=H_{2}\left(\mu_{1},\mu_{2}\right)$. Compute $c_{5}=c_{1}⊕(\mu_{1}\left|\right|\mu_{2})$, $c_{6}=(\mu_{1}\left|\right|\mu_{2})⊕H_{3}\left(e{\left(g,g\right)}^{r}\right)$, $c_{7}={\left(epk_{i}\right)}^{r\cdot H_{1}\left(epk_{i}\right)}$, $c_{8}=h^{r}$, $c_{9}=H_{4}{\left(c_{5},c_{6},c_{7},c_{8}\right)}^{r}$. At this point, secondary ciphertext is $C_{2}=\left(c_{2},c_{3},c_{4},c_{5},c_{6},c_{7},c_{8},c_{9}\right)$.

### 4.5. Security Equipment Redistribution and Ciphertext Update Phase

Once the user’s security equipment is stolen or lost, user needs to report to CA, then CA redistributes a security equipment to the user. Here, the work done by CA is similar to the previous security equipment distribution process, so it is omitted.

At the same time, CA also sends information to inform SS to update ciphertext. The information is as follows
(3)$$\begin{array}{cc} rk_{1} & =epk_{i}^{H_{1}\left(epk_{i}\right)\cdot {\left(z_{i}\cdot H_{1}\left(epk_{i}\right)\right)}^{-1}}\cdot h^{\epsilon }\\ rk_{2} & =epk_{i}^{H_{1}\left(epk_{i}\right)\cdot \epsilon } \end{array}$$
where $\epsilon \in_{R}Z_{q}^{*}$. After receiving $rk_{1},rk_{2}$, the SS updates the ciphertext $C_{2}$ as follows.
–Check
(4)$$\begin{array}{c} e\left(c_{7},h\right)=e\left(epk_{i}^{H_{1}\left(epk_{i}\right)},c_{8}\right)\\ e\left(c_{8},H_{4}\left(c_{5},c_{6},c_{7},c_{8}\right)\right)=e\left(h,c_{9}\right) \end{array}$$–If the above equations are not set up, the scheme stops. Otherwise it continues to execute.

Compute
(5)$$\begin{array}{cc} C_{10} & =\frac{e(c_{7},rk_{1})}{e(c_{8},rk_{2})}\\ & =\frac{e({\left(epk_{i}\right)}^{r\cdot H_{1}\left(epk_{i}\right)},epk_{i}^{H_{1}\left(epk_{i}\right)\cdot {\left(z_{i}\cdot H_{1}\left(epk_{i}\right)\right)}^{-1}}\cdot h^{\epsilon })}{e(h^{r},epk_{i}^{H_{1}\left(epk_{i}\right)\cdot \epsilon })}\\ & =\frac{e({\left(epk_{i}\right)}^{r\cdot H_{1}\left(epk_{i}\right)},h^{\epsilon })\cdot e({\left(epk_{i}\right)}^{r\cdot H_{1}\left(epk_{i}\right)},epk_{i}^{H_{1}\left(epk_{i}\right)\cdot {\left(z_{i}\cdot H_{1}\left(epk_{i}\right)\right)}^{-1}})}{e(h^{r},epk_{i}^{H_{1}\left(epk_{i}\right)\cdot \epsilon })}\\ & =e({\left(epk_{i}\right)}^{r\cdot H_{1}\left(epk_{i}\right)},epk_{i}^{H_{1}\left(epk_{i}\right)\cdot {\left(z_{i}\cdot H_{1}\left(epk_{i}\right)\right)}^{-1}})\\ & =e{\left(epk_{i},epk_{i}\right)}^{\frac{r\cdot H_{1}^{2}\left(epk_{i}\right)}{z_{i}\cdot H_{1}\left(epk_{i}\right))}}\\ & =e{\left(g^{z_{i}},g^{z_{i}}\right)}^{\frac{r\cdot H_{1}\left(epk_{i}\right)}{z_{i}}}\\ & =e{\left(g,g\right)}^{z_{i}\cdot r\cdot H_{1}\left(epk_{i}\right)} \end{array}$$

Finally, SS updates the ciphertext to $C_{2}^{*}=\left(c_{2},c_{3},c_{4},c_{5},c_{6},c_{10}\right)$.

### 4.6. Message Decryption

When users need to decrypt ciphertext, they can use security equipment and secret keys to decrypt. The two types of messages decryption are as follows:–Security equipment and ciphertext are not updated.Known $c_{5}=c_{1}⊕(\mu_{1}\left|\right|\mu_{2})$, so $c_{1}=c_{5}⊕(\mu_{1}\left|\right|\mu_{2})$. It is also known $c_{6}=(\mu_{1}\left|\right|\mu_{2})⊕H_{3}\left(e{\left(g,g\right)}^{r}\right)$. As a result, the following formula can be obtained
(6)$$\begin{array}{c} c_{1}=c_{5}⊕c_{6}⊕H_{3}\left(e{\left(g,g\right)}^{r}\right) \end{array}$$Because $c_{5}$ and $c_{6}$ are known, so users first use security equipment to compute $e{\left(g,g\right)}^{r}$. The process is as follows
(7)$$\begin{array}{cc} e{\left(g,g\right)}^{r} & =e(g,g^{r})\\ & =e(g,epk_{i}^{\frac{r}{z_{i}}})\\ & =e(g,epk_{i}^{r\cdot H_{1}\left(epk_{i}\right)\cdot \frac{1}{z_{i}\cdot H_{1}\left(epk_{i}\right)}})\\ & =e(g,c_{7}^{\frac{1}{z_{i}\cdot H_{1}\left(epk_{i}\right)}}) \end{array}$$By decryption of the user’s security devices, $c_{1}$ can be obtained. Next, CA checks whether the user’s *P* can satisfy *A* or not. If it is true, the CA computes $\tau_{A}=\prod \tau_{i}$, $\upsilon_{A}=\prod \upsilon_{i}$. The message can be decrypted as the following equation
(8)$$\begin{array}{c} m=\frac{c_{1}}{e\left(\tau_{A}\cdot \upsilon_{A},g^{k}\right)\cdot e\left(H_{4}\left(s\right),\beta_{A}^{k}\right)} \end{array}$$Finally, users verify the correctness of the message by checking whether $M=H_{4}\left(m\right)$. If $M=H_{4}\left(m\right)$ the computation result of the decryption, otherwise the message is wrong.–Security equipment and ciphertext had been updated. In this case, $e{\left(g,g\right)}^{r}$ can be calculated by the following formula
(9)$$\begin{array}{c} e{\left(g,g\right)}^{r}=c_{10}^{\frac{1}{z_{i}\cdot H_{1}\left(epk_{i}\right)}} \end{array}$$The rest of the decryption process is similar to the above, so it is omitted.

### 4.7. Correctness Verification

If a user’s attributes set *P* satisfies attribute sets *A* in specific access structure, the user is able to decrypt the message correctly. Therefore, we have that
(10)$$\begin{array}{c} \frac{c_{1}}{e\left(\tau_{A}\cdot \upsilon_{A},g^{k}\right)\cdot e\left(H_{4}\left(s\right),\beta_{A}^{k}\right)}\\ =\frac{m\cdot \alpha_{A}^{k}}{e\left(\tau_{A}\cdot \upsilon_{A},g^{k}\right)\cdot e\left(H_{4}\left(s\right),\beta_{A}^{k}\right)}\\ =\frac{m\cdot {\left(\prod \alpha_{i}\right)}^{k}}{e\left(\upsilon_{A}\cdot \prod \tau_{i},g^{k}\right)\cdot e\left(H_{4}\left(s\right),{\left(\prod \beta_{i}\right)}^{k}\right)}\\ =\frac{m\cdot {\left(\prod e{\left(H_{4}\left(s\right),g\right)}^{H_{2}\left(\alpha ||i\right)}\right)}^{k}}{e\left(\prod \left(H_{4}{\left(s\right)}^{H_{2}\left(\alpha ||i\right)}\right)\cdot \prod \left(H_{4}{\left(s\right)}^{-H_{2}\left(\beta ||i\right)}\right),g^{k}\right)\cdot e\left(H_{4}\left(s\right),{\left(\prod g^{H_{2}\left(\beta ||i\right)}\right)}^{k}\right)}\\ =\frac{m\cdot {\left(e{\left(H_{4}\left(s\right),g\right)}^{\sum H_{2}\left(\alpha ||i\right)}\right)}^{k}}{e{\left(\left(H_{4}{\left(s\right)}^{\sum H_{2}\left(\alpha ||i\right)}\right)\cdot \left(H_{4}{\left(s\right)}^{-\sum H_{2}\left(\beta ||i\right)}\right),g\right)}^{k}\cdot e{\left(H_{4}\left(s\right),\left(g^{\sum H_{2}\left(\beta ||i\right)}\right)\right)}^{k}}\\ =\frac{m\cdot {\left(e{\left(H_{4}\left(s\right),g\right)}^{\sum H_{2}\left(\alpha ||i\right)}\right)}^{k}}{e{\left(\left(H_{4}{\left(s\right)}^{\sum H_{2}\left(\alpha ||i\right)}\right)\cdot \left(H_{4}{\left(s\right)}^{-\sum H_{2}\left(\beta ||i\right)}\right),g\right)}^{k}\cdot e{\left(H_{4}{\left(s\right)}^{\sum H_{2}\left(\beta ||i\right)},\left(g\right)\right)}^{k}}\\ =\frac{m\cdot {\left(e{\left(H_{4}\left(s\right),g\right)}^{\sum H_{2}\left(\alpha ||i\right)}\right)}^{k}}{e{\left(\left(H_{4}{\left(s\right)}^{\sum H_{2}\left(\alpha ||i\right)}\right)\cdot \left(H_{4}{\left(s\right)}^{-\sum H_{2}\left(\beta ||i\right)}\right)\cdot \left(H_{4}{\left(s\right)}^{\sum H_{2}\left(\beta ||i\right)}\right),\left(g\right)\right)}^{k}}\\ =\frac{m\cdot {\left(e{\left(H_{4}\left(s\right),g\right)}^{\sum H_{2}\left(\alpha ||i\right)}\right)}^{k}}{e{\left(\left(H_{4}{\left(s\right)}^{\sum H_{2}\left(\alpha ||i\right)}\right),\left(g\right)\right)}^{k}}\\ =\frac{m\cdot {\left(e{\left(H_{4}\left(s\right),g\right)}^{\sum H_{2}\left(\alpha ||i\right)}\right)}^{k}}{{\left(e{\left(H_{4}\left(s\right),g\right)}^{\sum H_{2}\left(\alpha ||i\right)}\right)}^{k}}\\ =m \end{array}$$

## 5. Security Analysis

In this section, security analysis consistent with previous security models is given.

For the Type-1 security model, here an adversary A can get the user’s secret key $sk_{P}$, but it has no corresponding security equipment. Suppose A has got the secondary ciphertext $C_{2}=\left(c_{2},c_{3},c_{4},c_{5},c_{6},c_{7},c_{8},c_{9}\right)$ or updated ciphertext $C_{2}^{*}=\left(c_{2},c_{3},c_{4},c_{5},c_{6},c_{10}\right)$, which are all stored in SS, where $c_{2}=g^{k}$, $c_{3}=\beta_{A}^{k}$, $c_{4}=A$, $c_{5}=c_{1}⊕(\mu_{1}\left|\right|\mu_{2})$, $c_{6}=(\mu_{1}\left|\right|\mu_{2})⊕H_{3}\left(e{\left(g,g\right)}^{r}\right)$, $c_{7}={\left(epk_{i}\right)}^{r\cdot H_{1}\left(epk_{i}\right)}$, $c_{8}=h^{r}$, $c_{9}=H_{4}{\left(c_{5},c_{6},c_{7},c_{8}\right)}^{r}$, $c_{10}=e{\left(g,g\right)}^{z_{i}\cdot r\cdot H_{1}\left(epk_{i}\right)}$. A tries to compute $H_{3}\left(e{\left(g,g\right)}^{r}\right)$. Of course, $e{\left(g,g\right)}^{r}$ first needs to be calculated, that is
(11)$$\begin{array}{cc} e{\left(g,g\right)}^{r} & =e(g,g^{r})\\ & =e(g,epk_{i}^{\frac{r}{z_{i}}})\\ & =e(g,epk_{i}^{r\cdot H_{1}\left(epk_{i}\right)\cdot \frac{1}{z_{i}\cdot H_{1}\left(epk_{i}\right)}})\\ & =e(g,c_{7}^{\frac{1}{z_{i}\cdot H_{1}\left(epk_{i}\right)}}) \end{array}$$

From the above formula, it is easy to see that if A correctly guesses the $z_{i}$, it will be able to get $H_{3}\left(e{\left(g,g\right)}^{r}\right)$ successfully. Due to $z_{i}\in_{R}Z_{q}^{*}$, the probability of guessing is $\frac{1}{q}$. In addition, because of $H_{3}:G_{2}\to {\left\{0,1\right\}}^{*}$, A is able to correctly guess the output of $H_{3}$ with probability $\frac{1}{2^{*}}$. In summary, if A wants to access the correct message, the probability of its guess is $\frac{1}{2^{*}q}$. As long as *q* and ∗ are big enough, the probability is ignorable.

**Theorem** **1.**
*Suppose these hash function $H_{1},H_{2},H_{3}$, and $H_{4}$ are all random oracles. Our scheme is secure against chosen plaintext attack under the security model if the 1-wDBDHI assumption holds.*


**Proof** **of** **Theorem** **1.**If A just recovers the message with a secret key instead of a security equipment, then we can design an algorithm B to break 1-wDBDHI the assumption.*Setup.*B is assigned an example of 1-wDBDHI problem. B sets *g*, $y=g^{a}$, $Y=g^{b}$, $E=e{\left(g,g\right)}^{\frac{b}{a}}$, chooses $\psi ,\kappa \in_{R}Z_{q}^{*}$, and sends the public parameters param to be $\left(k,q,g,g_{1}=g^{\alpha },g_{2}=y,h=g^{\kappa },e\left(g,g\right),H_{1},H_{2},H_{3},H_{4}\right)$ to A. Among them, $H_{1},H_{2},H_{3},H_{4}$ are random oracles which is controlled by B.Phase 1. B receives the following queries from A.
Security equipment queries. B randomly chooses coin, where the value of coin is 0 or 1. Pr[coin=1]=ι.
When coin=0. B outputs b∈{0,1} randomly, and sends $\left(ID,epk_{i}=g^{z_{i}},coin=0,esk_{i}=z_{i}\right)$ to Equipmentlist.When coin=1. if $\left(ID,epk_{i},coin=1,z_{i}\right)$ has already existed in Equipmentlist. B sends $epk_{i}$ and $esk_{i}$ to A at the same time. Otherwise, B chooses $z_{i}\in_{R}Z_{q}^{*}$ and set $esk_{i}=z_{i}$, $epk_{i}=g^{z_{i}}$. Finally, B sends $\left(ID,epk_{i}=g^{z_{i}},coin=1,esk_{i}=z_{i}\right)$ to Equipmentlist, and returns $\left(epk_{i},esk_{i}\right)$ to A.Secret key queries. A sends a user’s ID to B, which queries the secret key of this user. B checks whether it has already owned ID and the corresponding $sk_{P}$. In this security model, B sends $sk_{P}$ to A.Message decryption queries. A sends a ciphertext to B. B decrypts the message as follows.For ciphertext $C_{2}$. B first checks whether there are tuples $\left(m,\mu_{1},\mu_{2},r\right)$, then sets $c_{7}={\left(epk_{i}\right)}^{r\cdot H_{1}\left(epk_{i}\right)}$, $c_{8}=h^{r}$, and $c_{9}=H_{4}{\left(c_{5},c_{6},c_{7},c_{8}\right)}^{r}$. Finally, B recovers $e{\left(g,g\right)}^{r}$ and computes $c_{1}=c_{5}⊕c_{6}⊕H_{3}\left(e{\left(g,g\right)}^{r}\right)$. For ciphertext $C_{2}^{*}$. The process of the query is similar to the above, so it is omitted.*Challenge.*A outputs $m_{0},m_{1}$ and $\left(ID^{*},epk_{i}^{*}\right)$. B gets $epk_{i}$ from the list EquipmentList. If $coin^{*}=1$, B aborts and outputs a,b∈{0,1}. Else, B proceeds.
For original ciphertext. B chooses $\mu_{1}^{*},\mu_{2}^{*}\in_{R}{\left\{0,1\right\}}^{*}$ and $b\in_{R}{\left\{0,1\right\}}^{*}$. It sets $c_{1}^{*}=m_{b}\cdot \alpha_{A}^{k}$, $c_{2}^{*}=g^{k}$, $c_{3}^{*}=\beta_{A}^{k}$, $c_{4}^{*}=A$, $c_{5}^{*}=c_{1}^{*}⊕(\mu_{1}^{*}\left|\right|\mu_{2}^{*})$, $c_{6}^{*}=(\mu_{1}^{*}\left|\right|\mu_{2}^{*})⊕H_{3}\left(E\right)$, $c_{7}^{*}={\left(Y\right)}^{z_{i}^{*}\cdot H_{1}\left(epk_{i}^{*}\right)}$, $c_{8}=Y^{\kappa }$, $c_{9}=Y^{H_{4}\left(c_{5}^{*},c_{6}^{*},c_{7}^{*},c_{8}^{*}\right)}$. B outputs $C_{2}^{**}=\left(c_{2}^{*},c_{3}^{*},c_{4}^{*},c_{5}^{*},c_{6}^{*},c_{7}^{*},c_{8}^{*},c_{9}^{*}\right)$.For updated ciphertext. B sets $C_{2}^{***}=\left(c_{2}^{*},c_{3}^{*},c_{4}^{*},c_{5}^{*},c_{6}^{*},c_{10}^{*}\right)$, here $c_{2}^{*}=g^{k}$, $c_{3}^{*}=\beta_{A}^{k}$, $c_{4}^{*}=A$, $c_{5}^{*}=c_{1}^{*}⊕(\mu_{1}^{*}\left|\right|\mu_{2}^{*})$, $c_{6}^{*}=(\mu_{1}^{*}\left|\right|\mu_{2}^{*})⊕H_{3}\left(E\right)$ and $c_{10}^{*}=e\left(g,Y^{z_{i}^{*}\cdot H_{1}\left(epk_{i}^{*}\right)}\right)$.Phase 2. A guesses a bit $b^{\prime }\in \left\{0,1\right\}$. These simulated $H_{1},H_{4}$ are perfect. If A does not either send $\mu_{1}^{*},\mu_{2}^{*}$ to $H_{2}$ or send *E* to $H_{3}$ before the challenge phase, the simulations of $H_{2}$ and $H_{3}$ also are perfect. $AskH_{2}^{*}$ and $AskH_{3}^{*}$ are denoted as events that $\left(\mu_{1}^{*},\mu_{2}^{*}\right)$ has been issued to $H_{2}$ and *T* has been issued to $H_{3}$, respectively.We assume that as long as B does not abort, the responses to the security equipment queries, secret key queries, and challenge phase are perfect. We denote Abort as the event that B aborts in the responses to the security equipment queries or in the challenge phase. Therefore, $Pr\left[⌝Abort\right]\geq \iota^{qse}\left(1-\iota \right)$. The maximum of ι is $\iota_{opt}=\frac{qse}{1+qse}$. Where qse is the total number of security equipment queries. Thus, the minimum of probability Pr[⌝Abort] is $\frac{1}{E\cdot (1+qse)}$, here E is the base of the natural logarithm.As long as A releases a valid original ciphertext with the help of $H_{2}$, the simulation of ciphertext update queries is also considered to be perfect. The probability of the error in these events is $Pr\left[CUE\right]\leq \frac{q_{cu}}{q}$, here $q_{cu}$ is the number of ciphertext update queries.As long as B does not refuse the queries of some valid ciphertexts, the simulation of message decryption queries is perfect. Val, $AskH_{2}$, and $AskH_{3}$ are these events that a valid ciphertext is returned, $\left(\mu_{1},\mu_{2}\right)$ is issued to $H_{2}$ and $e{\left(g,g\right)}^{r}$ is issued to $H_{3}$, respectively. According to the above simulation, we can get $Pr\left[Val|⌝AskH_{3}\right]\leq \frac{qH_{3}}{2^{l}}+\frac{1}{q}$ and $Pr\left[Val|⌝AskH_{2}\right]\leq \frac{qH_{2}}{2^{l}}+\frac{1}{q}$, here $qH_{3}$ and $qH_{2}$ are the numbers of querying $H_{2}$ and $H_{3}$, respectively. Pr[DRErr] is the probability that the event $Val|(⌝AskH_{2}\vee ⌝AskH_{3})$ occurs. Therefore, we can get $Pr\left[DRErr\right]\leq \left(\frac{qH_{2}\;+\;qH_{3}}{2^{l}}+\frac{2}{q}\right)\cdot qmd$, here qmd is the number of message decryption queries.Based on the analysis of the above three simulations, the following probability relationships can be calculated.
(12)$$\begin{array}{cc} \epsilon & =|Pr\left[b=b^{\prime }\right]-\frac{1}{2}|\\ & \leq \frac{1}{2}Pr\left[\left(H_{2}^{*}|⌝H_{3}^{*}\right)\vee H_{3}^{*}\vee CUE\vee DRErr|⌝Abort\right]\\ & \leq \frac{1}{2Pr[⌝Abort]}\left(AskH_{3}^{*}+\frac{qH_{2}+\left(qH_{2}+qH_{3}\right)qmd}{2^{l}}+\frac{2qmd+q_{cu}}{q}\right) \end{array}$$Therefore, we have that
(13)$$\begin{array}{cc} \epsilon^{\prime } & \geq \frac{1}{qH_{3}}\left(AskH_{3}^{*}\right)\\ & \geq \frac{1}{qH_{3}}\left(\frac{2\epsilon }{E(1+qse)}-\frac{qH_{2}+\left(qH_{2}+qH_{3}\right)qmd}{2^{l}}-\frac{2qmd+q_{cu}}{q}\right) \end{array}$$Finally, we have that
(14)$$\begin{array}{c} \left|Pr[A\left(g,g^{a},\ldots ,g^{a^{q}},g^{b},e{\left(g,g\right)}^{b/a}\right)=1\right]\\ -Pr\left[A\left(g,g^{a},\ldots ,g^{a^{q}},g^{b},e{\left(g,g\right)}^{z}\right)=1\right]|\geq \epsilon^{\prime } \end{array}$$
as required, which completes the proof. □

For the Type-2 security model, according to the previous construction, although an A has already owned security equipment, it has no secret key. Therefore, A does not have enough attributes to satisfy the *A*, and it can’t decrypt the ciphertext to get message. Because SS is a semi-trusted entity, message *m* is encrypted and uploaded by the DO. Meanwhile, attribute sets and access policy formulated by DO are also incorporated into ciphertext.

In the process of ABE, a message *m* is obfuscated with the $\alpha_{A}^{k}$, here, $\alpha_{A}$ is determined by the attribute sets and access policy *A*, the *k* is security parameter which is randomly generated in every oracle. From the above analysis, the $\alpha_{A}^{k}$ is secure, and the message *m* is also secure.

In the process of decryption, A does not have $sk_{P}$ or has the wrong secret key, it can’t get the attribute set *P* which is granted by CA. CA is a trusted party, it won’t have a collusion attack with SS. From the above analysis, as long as $\alpha_{A}^{k}$ is not easy to crack, the message *m* is secure.

## 6. Performance Analysis

In this section, the performance of proposed scheme is analyzed from different perspectives. Meanwhile, the comparison with [14,38] is also analyzed in terms of features, communication and computational cost. The results of the comparison reveal that our scheme is more suitable for the protection of public data resources and achieves more functions, but dosen’t require a great increase of cost. In general, our scheme is more suitable to be practically deployed

First of all, some notations used in efficiency analysis are defined as follows. $|G_{1}|$ and $|G_{2}|$ are utilized to denote the length of an element in groups $G_{1}$ and $G_{2}$, *l* denotes the length of security parameter, ck denotes the key length of a symmetric encryption algorithm, $|Z_{q}^{*}|$ denotes the length of an element in $Z_{q}^{*}$. |m| and |∗| denote the length of message *m* and arbitrary 01 strings, respectively. PA, $EXP_{1}$, $EXP_{2}$, and *H* are utilized to denote the cost of a bilinear pairing, an exponentiation in $G_{1}$, an exponentiation in $G_{2}$ and a one-way hash function, respectively. EM and DM are utilized to denote the cost of symmetric encryption and decryption, respectively. In our scheme, the calculation of the bilinear pairing and the exponentiation is over the supersingular elliptic curve, which is defined in preliminaries. Thus, the computational complexities of the bilinear pairing is $O(m^{2})$. Here, *m* is the extension degree of the finite field $Z_{q}^{*}$. *n* denotes the number of users.

For features comparison. We compare our scheme with [14,38] in terms of access control policy, two-factor protection, equipment revocation and key and equipment separation. The results are shown in Table 2. Compared with [14,38] respectively, it can be seen that only our scheme can achieve all the three functions at the same time. Compared with our scheme, ref. [14] cannot simultaneously share data with a group of users, limited scope of application may be a major drawback. Especially, it is worth saying that since [38] does not support the two-factor mechanism, which greatly reduces the security of this scheme. In short, refs. [14,38] only achieve partial design goal of our scheme.

For theoretical comparison. Communication, computational and time complexity comparison are demonstrated in Table 3, Table 4 and Table 5, respectively. In Table 4, KSED, PE, SE, SERCU, MDSC, and MDUC are initial capitalization of keygen and security equipment distribution, primary encryption, secondary encryption, security equipment redistribution and ciphertext update, message decryption (from secondary ciphertext) and message decryption (from updated ciphertext), respectively. In the rest of this paper, this abbreviation will also be used. Compared with [38] which lacks two-factor data protection, it can be seen that our scheme requires increase a little computational cost in security equipment redistribution and ciphertext update phase. This is because the redistribution of security equipment is supported by our scheme. In the process of ciphertext generation, our scheme does not require a symmetric encryption operation. Furthermore, it is worth of mentioning that cost of secondary encryption can be outsourced to SS. A similar situation also exists in Table 3, which is that our scheme needs extra communication cost in transmission of security equipment and updating ciphertext. However, the total cost of our scheme is less than [38]. Compared with [14] which lacks public data protection, the cost of our scheme is less than [14] at all phases, this is mainly due to the fact that an efficient ABE is adopted by our scheme. In addition, under the premise of ensuring safety, the generation and redistribution of security equipment of our scheme is also more streamlined than [14]. It can be seen from Table 5 that the time complexity of our scheme is not higher than [14,38]. Moreover, the computational complexity of our scheme is linearly related with the number of users, which indicates that our scheme is suitable for a real time security guard scenario.

For practical comparison. In practical efficiency test, the test environment is set to be: Intel(R) Core(TM) i5-6500 CPU @ 3.2 GHz, 8 GB RAM, GNU Multiple Precision Arithmetic (GMP) library, Pairing-Based Cryptography (PBC) library, and C language are used on a Linux system with Ubuntu 16.04 TLS. Microsoft Office Excel 2016 and Matlab 2016a are used by us as tools for drawing statistical figures. As we all know, PBC library is a free and portable C language library. Through an abstract interface, programmers can implement pair-based cryptosystem without considering the specific mathematical details or even the knowledge of elliptic curve and theory. Ubuntu 16.04 TLS is a free open source desktop operating system based on Linux, which combines Windows visualization and Linux stability. The comparison results are shown in Figure 3, as the number of users increases, the computational cost of the three schemes almost increases linearly, but the growth trend of our scheme is slow, and it is consistent with previous theoretical analysis.

In order to make a more comprehensive comparison, the computational cost of attributes is also introduced. As shown in Figure 4, (a) is the three-dimensional experimental result of our scheme, (b) is the experimental result of [38], and (c) is the experimental result of [14]. We set the number of users varies from 0 to 10,000 and the number of attributes varies from 0 to 1,000,000. The computational cost increases when either the number of users or the number of attributes. It is easy to see that although upward trends of the three figures are similar, the trend of our scheme is more gentle. For example, when the number of users and the number of attributes are 1000 and 1,000,000 respectively, time less than 600 s is consumed in our scheme, whereas [14,38] take more than 600 s, which also indicates the high efficiency of our scheme.

Figure 5 compares computational cost of our scheme with [14,38] from different phases. Here, the number of attributes per user is set to 2. (a) shows the computational time for generating key and distributing security equipment. Apparently, the computational cost of [14] is a little more than our scheme, however, the computational cost of [38] is much greater than our scheme and grows faster. The main reason for this situation is that there are more bilinear pairing operations and hash functions in [14,38] at this phase. (c), (d) and (f) demonstrate the computational time of secondary encryption, security equipment redistribution, and ciphertext update and message decryption (from updated ciphertext) respectively. The analysis of them is similar to (a) and is not covered here. (b), and (e) show the computational cost of primary encryption and message decryption (from secondary ciphertext) respectively. In the above two phases, the computational cost of [38] is less than our scheme. The main reason is that symmetric encryption and decryption algorithms are applied in [38] at these two phases. It is well known that symmetric encryption algorithms are generally more efficient than public key encryption algorithms. However, the total computational cost of each phase of [38] is still higher than our scheme.

In order to evaluate the performance more intuitively, the simulation experiment of our scheme, refs. [14,38] are deployed on a mobile device. The experiment is implemented on Nexus 5X Android virtual machine with Four-core CPU, 2 GB running memory, and 32 G body memory. Moreover, the codes are written in Android Studio with Java programming language to obtain the experimental results. Figure 6 is one of experimental results of three different schemes obtained from the mobile device. It is easy to see from Figure 6 that 1607.48 ms, 2221.67 ms, and 1977.32 ms are cost of our schme, [14,38] respectively when the number of attributes is 50. The simulation result for each phase is also consistent with Figure 5. Obviously, our scheme is more suitable for deployment in a real-world application.

In short, through the above various performance analysis, it is not difficult to find that our scheme can achieve two-factor protection for public data and redistribution of security equipment with less overhead.

## 7. Conclusions

Various data resources are important objects of ocean management, which may involve navigation, mining, shipping, and even national security, secure and efficient data protection schemes are especially needed. At present, the research on two-factor data security protection scheme and smart ocean management is flourishing. Given the shortcomings of existing schemes, we propose a two-factor-based public data protection scheme in smart ocean management. In our scheme, DOs are allowed to encrypt the message with some attributes (including access policy), users who satisfy certain attributes can combine their own secret keys and security equipment to decrypt ciphertext. In addition, the revocation of security equipment is also an important advantage of our scheme, which solves the problem of equipment loss well and brings many conveniences to smart ocean management. The analysis of security and performance shows that our scheme is more efficient than similar schemes on the premise of ensuring security.

Although sufficient contributions have been included in our works, there are still several challenges that we will leave as a future work. Firstly, our scheme supports one-time equipment revocation that may be not sufficient enough in practice. Therefore, multiple revocation for equipment will be one of the next works to be completed. Secondly, the access policy in our scheme is only an abstract framework and does not involve specific access control structures, such as tree or matrix. Therefore, how to design a specific access control structure to better adapt to smart ocean management is one of the further works.

## Figures and Tables

**Figure 1 sensors-19-00129-f001:**
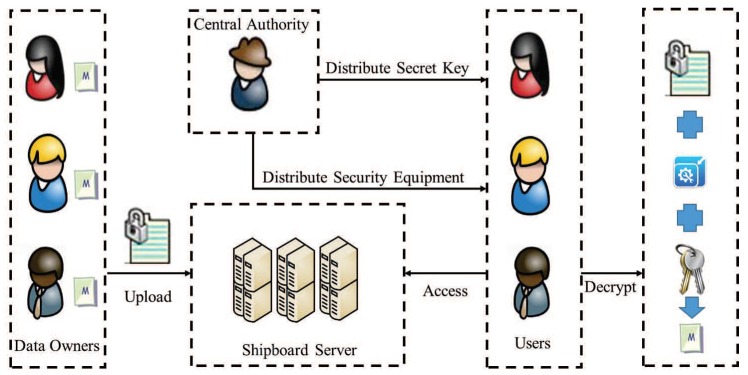
The system model.

**Figure 2 sensors-19-00129-f002:**
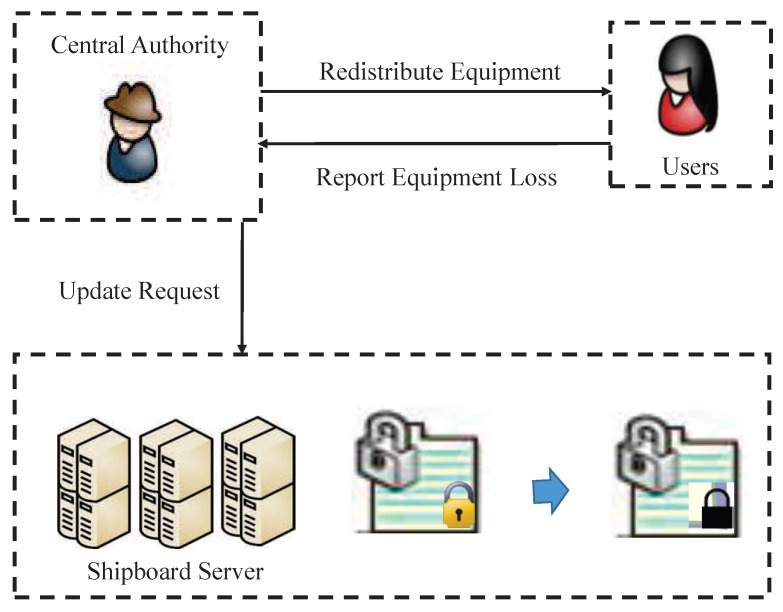
The process of ciphertext update and security equipment redistribution.

**Figure 3 sensors-19-00129-f003:**
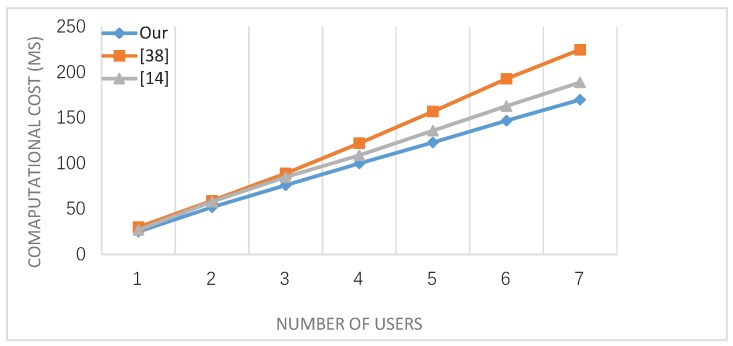
The computational cost comparison.

**Figure 4 sensors-19-00129-f004:**
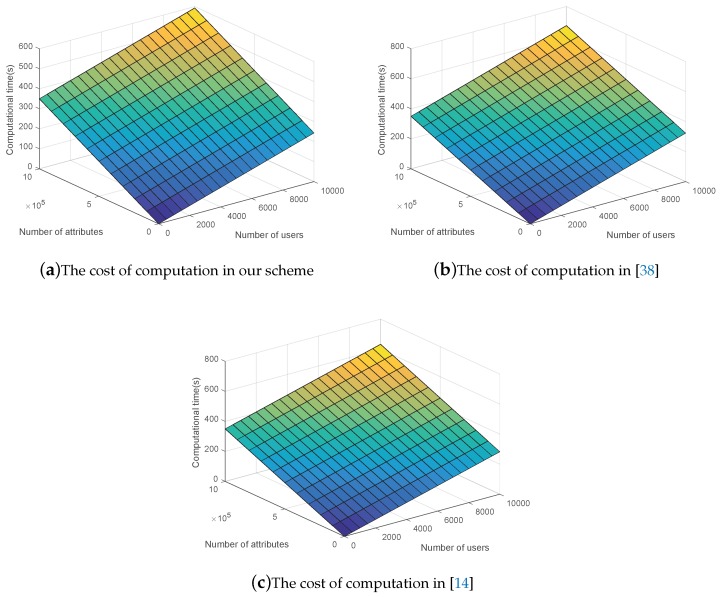
The computational cost comparision.

**Figure 5 sensors-19-00129-f005:**
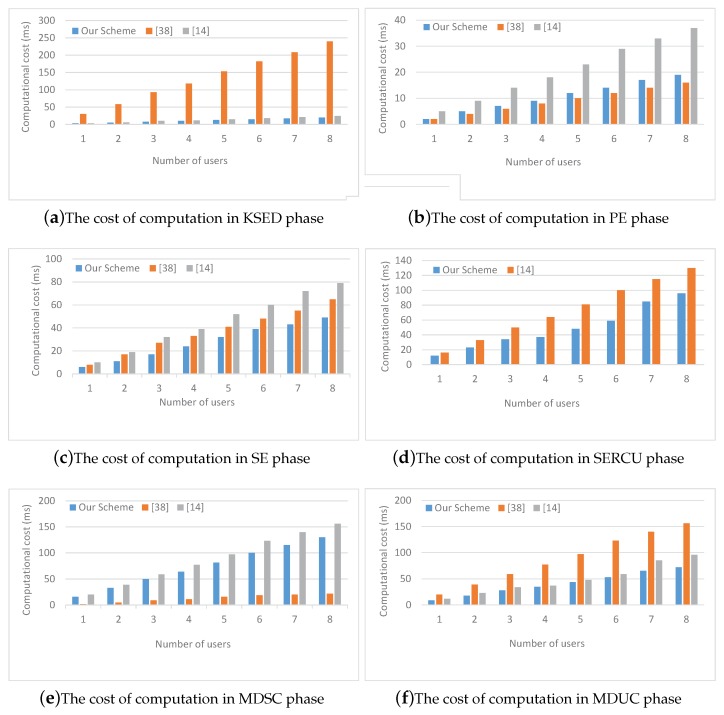
The computational cost comparision in different phases.

**Figure 6 sensors-19-00129-f006:**
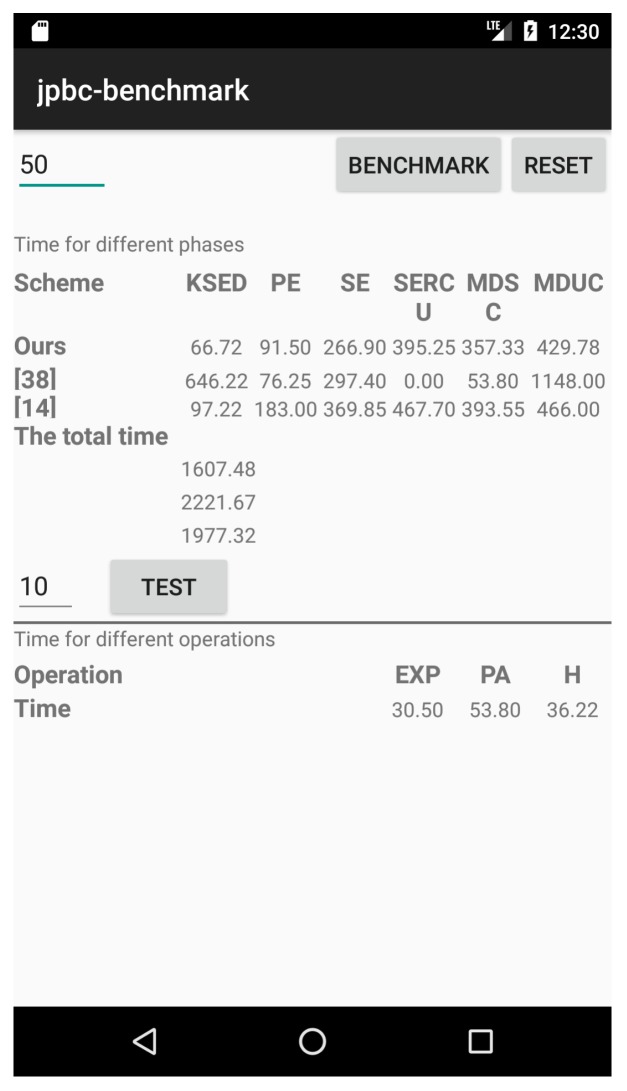
The experimental results of different phases on mobile device.

**Table 1 sensors-19-00129-t001:** Notations.

Notation	Interpretation
$G_{1}$, $G_{2}$	cyclic multiplicative groups
*p*	prime order of $G_{1}$, $G_{2}$
*g*, $g_{2}$, *h*	generators of $G_{1}$
$Z_{q}^{*}$	set of nonnegative integers less than *q*
$H_{1}$, $H_{2}$, $H_{3}$, $H_{4}$	one-way hash function
*m*	a message
⊕	exclusive OR
$ID_{i}$	the identity of user *i*
*A*	the attribute set (including access policy)
*V*	attributes value
*P*	the attribute that users possess
$C_{1}$	the primary ciphertext
$C_{2}$	the secondary ciphertext
$C_{2}^{*}$	updated ciphertext
$epk_{i}$	the public information of security equipment
$esk_{i}$	the secret information of security equipment
$sk_{P}$	the secret key of users

**Table 2 sensors-19-00129-t002:** Features comparison of related schemes.

Schemes	Access Control Policy	Two Factors	Equipment Revocation	Key and Equipment Separation
Ours	ABE	Yes	Yes	Yes
[38]	ABE	No	No	Yes
[14]	IBE	Yes	Yes	No

**Table 3 sensors-19-00129-t003:** Communication cost comparison.

Schemes	Ours	[38]	[14]
secret key length	$2|G_{1}|$	$3|Z_{q}^{*}|$	$2|G_{1}|$
security equipment length	$|G_{1}|$ + $|Z_{q}^{*}|$	⊥	$2|G_{1}|$ + $2|Z_{q}^{*}|$
primary ciphertext length	$\left(|m\right|+2)|G_{1}|$	|ck|	$6|G_{1}|$ + 4l
secondary ciphertext length	$\left(|m\right|+5)|G_{1}\left|+|*\right|$	$|G_{1}\left|+\right|G_{2}|$	$3|G_{1}|$ + $|G_{2}|$ + 4l
updated ciphertext length	$\left(|m\right|+2)|G_{1}\left|+|*|+\right|G_{2}|$	⊥	$2|G_{1}|$

**Table 4 sensors-19-00129-t004:** Computational cost comparison.

Phases	Ours	[38]	[14]
KSED	$EXP_{1}+4H$	$12EXP_{1}+2EXP_{2}+H$	$4EXP_{1}$
PE	$3EXP_{1}$	EM	$2EXP_{1}+EXP_{2}+PA+3H$
SE	$3EXP_{1}+EXP_{2}+4H$	$4EXP_{1}+EXP_{2}+2H$	$3EXP_{1}+EXP_{2}+PA+3H$
SERCU	$4EXP_{1}+6PA+5H$	⊥	$6EXP_{1}+6PA+5H$
MDSC	$7EXP_{1}+2PA+3H$	$PA+2EXP_{2}$	$9EXP_{1}+2PA+3H$
MDUC	$6EXP_{1}+EXP_{2}+2PA+3H$	DM	$8EXP_{1}+EXP_{2}+2PA+2H$

**Table 5 sensors-19-00129-t005:** Time complexity comparison.

Time Complexity	Ours	[38]	[14]
communication complexity	$O(n^{2})$	$O(n^{2}\sqrt{n})$	$O(n^{2})$
computational complexity	$O(nm^{2})$	$O(nm^{2})$	$O(nm^{2})$

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
