# Peer review of "Two-Factor-Based Public Data Protection Scheme in Smart Ocean Management"

_sensors, 2019, doi:10.3390/s19010129_

Reviewer 1 Report

In this paper, authors focused on the study of two-factor-based public data protection in smart ocean management. Authors developed an attribute-based encryption (ABE) data protection scheme and a two-factor-based data protection method which provides revocation in smart ocean management. Basically, the motivation of the paper is solid as well as the proposed scheme and solution are interesting and novel. Moreover, authors evaluated the performance of the proposed scheme with detailed analysis. Furthermore, I do like the organization of the paper and presentation quality of the manuscript so that readers are able to follow the manuscript well.
To improve the paper, I have the below minor requests to be modified by authors.
1. It is recommended that authors will do investigate more critical related works and will add those works into the manuscript. Besides, it is recommended that authors will find important related works which were published in MDPI Sensors. For example, the below related works can be considered as References.
- G. Xu, S. Qiu, H. Ahmad, G. Xu, Y. Guo, M. Zhang, H. Xu, “A Multi-Server Two-Factor Authentication Scheme with Un-Traceability using Elliptic Curve Cryptography”, MDPI Sensors, 2018. https://doi.org/10.3390/s18072394
- H. Kim, J. Ben-Othman, “A Collision-Free Surveillance System Using Smart UAVs in Multi Domain IoT”, IEEE Communications Letters, pp. 2587-2590, December, 2018.
- K. Lim, K. Tuladhar, H. Kim, “Detecting Location Spoofing using ADAS Sensors in VANETS”, Proceedings of IEEE CCNC, January, 2019.
2. It is recommended that in Section 6, authors will explain system setting, tool, the reason why authors utilized those settings with possible additional information which will allow readers to follow the manuscript reasonably.
3. It is recommended that authors suggest future research challenges and works for readers in Section 7.

Author Response

We would like to thank the reviewer for his/her insightful comments on our paper. We have addressed all of the reviewer's concerns/comments and revised the manuscript to the best of our ability according to the comments provided by the reviewer.
Description: In this paper, authors focused on the study of two-factor-based public data protection in smart ocean management. Authors developed an attribute-based encryption (ABE) data protection scheme and a two-factor-based data protection method which provides revocation in smart ocean management. Basically, the motivation of the paper is solid as well as the proposed scheme and solution are interesting and novel. Moreover, authors evaluated the performance of the proposed scheme with detailed analysis. Furthermore, I do like the organization of the paper and presentation quality of the manuscript so that readers are able to follow the manuscript well.
Point 1: It is recommended that authors will do investigate more critical related works and will add those works into the manuscript. Besides, it is recommended that authors will find important related works which were published in MDPI Sensors. For example, the below related works can be considered as References.

- G. Xu, S. Qiu, H. Ahmad, G. Xu, Y. Guo, M. Zhang, H. Xu, “A Multi-Server Two-Factor Authentication Scheme with Un-Traceability using Elliptic Curve Cryptography”, MDPI Sensors, 2018. https://doi.org/10.3390/s18072394
- H. Kim, J. Ben-Othman, “A Collision-Free Surveillance System Using Smart UAVs in Multi Domain IoT”, IEEE Communications Letters, pp. 2587-2590, December, 2018.
- K. Lim, K. Tuladhar, H. Kim, “Detecting Location Spoofing using ADAS Sensors in VANETS”, Proceedings of IEEE CCNC, January, 2019.
Response 1: Thank you for reviewer's helpful comment. The related works recommended by the reviewer have been added to the references.  We have marked it in red font in the revised manuscript.
Point 2: It is recommended that in Section 6, authors will explain system setting, tool, the reason why authors utilized those settings with possible additional information which will allow readers to follow the manuscript reasonably.
Response 2: Thank you for reviewer's valuable comment. We have explained system setting, tool and the reasons for using them. The revised explanations are shown as follows:
“For practical comparison. In practical efficiency test, the test environment is set to be: Intel(R) Core(TM) i5-6500 CPU @ 3.2 GHz, 8 GB RAM, GNU Multiple Precision Arithmetic (GMP) library, Pairing-Based Cryptography (PBC) library and C language are used on a Linux system with Ubuntu 16.04 TLS. Microsoft Office Excel 2016 and Matlab 2016a are used by us as tools for drawing statistical figures. As we all know, PBC library is a free and portable C language library. Through an abstract interface, programmers can implement pair-based cryptosystem without considering the specific mathematical details or even the knowledge of elliptic curve and theory. Ubuntu 16.04 TLS is a free open source desktop operating system based on Linux, which combines Windows visualization and Linux stability.”
Point 3: It is recommended that authors suggest future research challenges and works for readers in Section 7.
Response 3: Thank you for reviewer's valuable comment.  Future research challenges and works have been added to Section 7. The following paragraph has been added as research challenges and works in the revised manuscript.
“Although sufficient contributions have been included in our works, there are still several challenges that we will leave as a future work. Firstly, our scheme supports one-time equipment revocation that may be not sufficient enough in practice. Therefore, multiple revocation for equipment will be one of the next works to be completed. Secondly, the access policy in our scheme is only an abstract framework and does not involve specific access control structures, such as tree or matrix. Therefore, how to design a specific access control structure to better adapt to smart ocean management is one of the further works.”
We hope that our explanations have satisfied the reviewer’s concerns.
Again, we really appreciate the reviewer’s valuable and helpful comments on our paper. Thank you.

Reviewer 2 Report

The authors study the problem of two-factor data security protection in the smart ocean management. Specifically, they propose a two-factor-based protection scheme with factor revocability in smart ocean management, where the data owner sends encrypted messages to users through a shipboard server and is required to formulate access policies and perform attribute-based encryption on messages. On the users’ side, a two-factor decryption is performed with a security key at the first level and through the security equipment, i.e., sensor card, at the second level. The proposed research work is somehow interesting as it addresses up to a level the security issues in the smart ocean management setup, however, the proposed two-factor data security by itself is not novel, as it has already been proposed in the authentication systems. The authors should better clarify their own contributions in section 1.1 in terms of fundamental differences both in the theoretical and analytical framework, as well as in the application layer. Figures 1 and 2 should be resized as currently they are huge. My main concern as it is raised based on the provided analysis in sections 4 and 5, is the realistic applicability of the proposed two-factor data security framework. What is the implementation cost of the proposed framework? What is the time complexity of the proposed framework, so as to be implemented in a real time security guard scenario? The results are sufficient in order to support the main attributes of the proposed framework. The overall manuscript should be checked regarding the usage of the English language, as it has many typos, grammar and syntax errors.

Author Response

We would like to thank the reviewer for his/her insightful comments on our paper. We have addressed all of the reviewer's concerns/comments and revised the manuscript to the best of our ability according to the comments provided by the reviewer.
Description: The authors study the problem of two-factor data security protection in the smart ocean management. Specifically, they propose a two-factor-based protection scheme with factor revocability in smart ocean management, where the data owner sends encrypted messages to users through a shipboard server and is required to formulate access policies and perform attribute-based encryption on messages. On the users’ side, a two-factor decryption is performed with a security key at the first level and through the security equipment, i.e., sensor card, at the second level. The proposed research work is somehow interesting as it addresses up to a level the security issues in the smart ocean management setup.
Point 1: The proposed two-factor data security by itself is not novel, as it has already been proposed in the authentication systems.
Response 1: Thank you for reviewer's helpful comment. As the reviewer said, the two-factor data security has been widely used in the authentication systems and it is not novel. However, two-factor is only a structure that can be implemented by a variety of technologies and applied to different cryptographic schemes.
We take the paper “Guosheng X, Shuming Q, Haseeb A, et al. A Multi-Server Two-Factor Authentication Scheme with Un-Traceability Using Elliptic Curve Cryptography[J]. Sensors, 2018, DOI:10.3390/s18072394.” as an example. In the works of Guo et al., authentication-and-key-agreement are regarded as two factors to design an authentication scheme which uses elliptic curve cryptography.  However, ciphertext policy attribute-based encryption and public key encryption are considered as two factors in our scheme. Therefore, the two factors are indeed an old concept, but it is also a classic concept and has great research value.
In order to make the logic of our scheme more rigorous, the following paragraph has been added in the revised manuscript.
“We note that there are some other research works on two factors, such as \cite{Xu2018A}. Nevertheless, the two factors in their works are different from ours, they leverage different techniques to achieve different systems. Here, we will not compare them with our present scheme.”
Point 2: The authors should better clarify their own contributions in section 1.1 in terms of fundamental differences both in the theoretical and analytical framework, as well as in the application layer.
Response 2: Thank you for reviewer's helpful comment. We have checked the contributions of our paper according to the reviewer's comment. The revised contributions are listed as follows:
A practical attribute-based encryption (ABE) data protection scheme is proposed. In practical applications, public resources are more likely to be shared by many users. The security of these public resources is of great significance. In our scheme, we take advantage of an efficient ABE to address this issue. Ensure that only users who satisfy specific attributes can access public data.
A two-factor-based data protection scheme that supports revocation in smart ocean management is proposed. We apply this kind of data protection scheme to smart ocean management for the first time. Ocean-related data are often highly confidential, however, single factor scheme does not meet the security requirements of the application layer. Therefore, we design two-factor-based data protection scheme. In addition, when performing missions in the ocean environment, the sensor cards of the crew will inevitably be lost, for example slipping into the sea. Here, the revocation of the sensor cards is particularly important. Therefore, the revocation and redistribution of security equipment is also supported by our scheme.
More formal and complete security and performance analysis are proposed. According to the designed security model, descriptive language and rigorous mathematical proof are used in security analysis. The attacker's success is reduced to the resolution of difficult problems. The security of the scheme is proved probabilistically. In performance analysis, similar schemes are compared with ours from different phases, thus achieving a more fine-grained comparison.
Point 3: Figures 1 and 2 should be resized as currently they are huge.
Response 3: Thank you for your helpful comment. The sizes of Figures 1 and 2 have been adjusted appropriately.
Point 4: My main concern as it is raised based on the provided analysis in sections 4 and 5, is the realistic applicability of the proposed two-factor data security framework. What is the implementation cost of the proposed framework
Response 4: Thank you for reviewer's valuable comment. Our main work is to conduct theoretical research and simulation experiments. The results of nalysis and experiment show that our scheme has better realistic applicability than the existing similar schemes. As for the final implementation and cost, it needs to be further discussed with engineers and financial personnel. However, we believe that our secure and efficient scheme will provide a solid theoretical support for the specific implementation.
Point 5: What is the time complexity of the proposed framework, so as to be implemented in a real time security guard scenario?
Response 5: Thank you for reviewer's valuable comment. Time complexity is divided into communication complexity and computational complexity, which has been added to the revised manuscript and can be seen in Section 6 and Table 5.
Point 6: The overall manuscript should be checked regarding the usage of the English language, as it has many typos, grammar and syntax errors.
Response 6:  Thank you for your helpful comment.  We have been checked regarding the usage of the English language and corrected typos, grammar and syntax errors in the revised manuscript.
We hope that our explanations have satisfied the reviewer’s concerns.
Again, we really appreciate the reviewer’s valuable and helpful comments on our paper. Thank you.

Round  2

Reviewer 2 Report

The authors have successfully addressed all the comments pointed by the reviewer. There are no major concerns regarding this manuscript.